# Most endovascular thrombectomy patients have Target Mismatch despite absence of formal CT perfusion selection criteria

Jayden C. Subramaniam[1], Andrew Cheung[2,3], Nathan Manning[2,3], Justin Whitley[3], Dennis Cordato[4,5], Alessandro Zagami[6], Cecilia Cappelen-Smith[4,5], Huiqiao Tian[1], Chris Levi[7], Mark Parsons[1,5], Ken S. Butcher[1,6]*

1 Faculty of Medicine, University of New South Wales, Sydney, Australia, 2 Department of Interventional Neuroradiology, The Prince of Wales Hospital, Sydney, Australia, 3 Department of Interventional Neuroradiology, Liverpool Hospital, Liverpool, Australia, 4 Department of Neurology and Neurophysiology, Liverpool Hospital, Liverpool, Australia, 5 Ingham Institute of Applied Medical Research, Sydney, Australia, 6 Institute of Neurological Sciences, Prince of Wales Hospital, Sydney, Australia, 7 Departments of Neurology, John Hunter Hospital, University of Newcastle, Newcastle, Australia

* ken.butcher@unsw.edu.au

**Data Availability Statement:** All relevant data are within the manuscript and its Supporting Information files.

## Abstract

Endovascular thrombectomy (EVT) is the standard of care for large vessel occlusion stroke. Use of Computed Tomographic Perfusion (CTP) to select EVT candidates is variable. The frequency of treatment and outcome in patients with unfavourable CTP patterns is unknown. A retrospective analysis of CTP utilisation prior to EVT was conducted. All CTP data were analysed centrally and a Target Mismatch was defined as an infarct core ≤70 ml, penumbral volume ≥15ml, and a total hypoperfused volume:core volume ratio >1.8. The primary outcome was good functional outcome at 90 days, defined as a modified Rankin Scale (mRS) score 0–2. follow-up infarct volume, core expansion and penumbral salvage volumes were secondary outcomes. Of 572 anterior circulation EVT patients, CTP source image data required to generate objective maps were available in 170, and a Target Mismatch was present in 151 (89%). The rate of 90-day good functional outcome was similar between Target Mismatch (53%) and Large Core Non-Mismatch groups (46%, p = 0.629). Median follow-up infarct volume in the Large Core Non-Mismatch group (104ml [IQR 25ml-189ml]) was larger than that in the Target Mismatch patients (16ml [8ml-47ml], p<0.001). Despite a lack of formal CTP selection criteria, the majority of patients treated at our centres had a Target Mismatch. Patients without Target Mismatch had larger follow-up infarct volumes, but the functional recovery rate was similar to that in Target Mismatch patients. Infarct volumes should be included as objective assessment criteria in the evaluation of the efficacy of EVT in non-Target Mismatch patients.

**Funding:** KB is supported by the NSW Health 18/19 Senior Cardiovascular Scientist award. Grant number: RG192264. Name of funder: NSW Health (https://www.health.nsw.gov.au/). The funder had no role in study design, data collection and analysis, decision to publish, or preparation of the manuscript.

**Competing interests:** The authors have declared that no competing interests exist.

## Introduction

The standard of care for appropriately selected ischaemic stroke patients with a large vessel occlusion (LVO) is endovascular thrombectomy (EVT) [1]. Patients presenting within 6 hours of symptom onset have been shown to experience improved functional outcomes when treated with EVT compared to medical care [2]. EVT has also been demonstrated to be superior to medical care in LVO patients, selected using specific perfusion imaging criteria 6–24 hours after onset [3, 4].

EVT benefit in the late window (6–24 hours after onset) has been demonstrated in patients with evidence of potentially salvageable tissue, based on advanced Computed Tomographic (CT) or Magnetic Resonance (MR) imaging techniques [3–5]. The most common patient selection method utilised is CT Perfusion (CTP) based estimates of acute infarct core and ischaemic penumbra. The latter can be quantified with post-processing of raw image data using the RAPID software package to provide volumes of estimated core and penumbra [3–5]. While the improved outcomes in this group of highly selected patients is unequivocal, an EVT treatment benefit in patients selected without quantitative CTP has not been excluded. Furthermore, the utility of CTP in selection of EVT candidates within 6 hours of onset, is unproven [6]. Systems of care have not kept pace with evidence in all centres, resulting in variable access to CTP acquisition as well as objective post-processing of raw perfusion data.

Current patient workup and selection criteria for EVT varies between centres treating patients who present with LVO. The extent to which extrapolation from trial evidence occurs in practice is unknown. The effect of treatment in patients selected outside narrow trial inclusion/exclusion criteria is also unknown. We aimed to determine the frequency of EVT treatment in patients without Target Mismatch, determined with quantitative CTP, at two comprehensive stroke centres. We tested the hypothesis that the frequency of good outcomes in patients with unfavourable CTP imaging patterns would be lower than that in patients with Target Mismatch, despite EVT.

## Methods

### Centres

The University of New South Wales has two affiliated comprehensive stroke centres where EVT is offered. CTP is performed routinely at both centres and raw images are post-processed with the RAPID (Rapid Processing of Perfusion and Diffusion, iSchemaView Inc, Menlo Park, California, United States) or MIStar (Apollo Medical Imaging Technology, Melbourne, Victoria, Australia) software packages. Approximately, half of the EVT patients are transferred from one of 9 referring primary stroke centres, where CTP acquisition is variable, with even more limited access to objective post-processing software (RAPID/MIStar). Patient selection is at the discretion of the treating interventional neuroradiologists (AC, NM, JW), who consider all imaging data available, but do not employ formal trial inclusion/exclusion criteria, particularly with respect to quantitative CTP measurements.

### Patients

All patients presented with acute anterior circulation ischaemic stroke associated with a Computed Tomographic Angiography (CTA) proven LVO and subsequently underwent EVT between 1st January 2018 and 31st March 2020. Clinical data were all collected in a prospective database. Patients were excluded from the final analysis if they did not have a pre-treatment CTP or source images were not archived (i.e. only subjective hardware vendor maps were available). All patients underwent EVT with aspiration and/or stent retriever devices. This

analysis was approved by the Human Research Ethics committees at both comprehensive stroke centres: South Western Sydney Local Health District Human Research Ethics Committee (approval number: LNR/15/LPOOL/479) and the South Eastern Sydney Local Health District Human Research Ethics Committee (approval number: 2019/ETH11534). The Ethics Committees waived the requirement for informed consent and data were not anonymised before being accessed.

### Image acquisition

All Computed Tomography (CT) images were acquired using site-specific protocols at the presenting hospital. A whole brain non-contrast CT (NCCT), as well as extracranial and intracranial CTA and CTP were obtained in all patients at the time of the initial assessment. Images were acquired on GE Medical Systems Revolution, Siemens SOMATOM Definition AS+, Siemens SOMATOM Force, Phillips IQon Spectral CT and Toshiba/Canon Aquilion ONE scanners.

### Clinical and radiographic assessment

Baseline National Institute of Health Stroke Scale (NIHSS) scores were assessed by members of the treating clinical team at the time of presentation. Recanalisation was rated by the treating interventional neuroradiologist, using the modified Thrombolysis in Cerebral Infarction (mTICI) scale score at the conclusion of EVT. Follow-up imaging (CT and Magnetic Resonance Imaging (MRI), including Diffusion-Weighted Imaging (DWI) sequences) were acquired as part of clinical care between 12 hours and 2 weeks after EVT. Day 90 modified Rankin Scale (mRS) scores were assessed by members of the treating clinical team (not blinded to treatment) during routine clinical follow-up visits, which were conducted by telephone in some cases. The time from symptom onset to CTP was calculated using the contemporaneously maintained clinical database. Patients with a time to CTP greater than 6 hours were classified as late window presenters and were the subject of a secondary analysis.

### Image analysis

All CTP source images were centrally post-processed using MIStar software. Quantitative maps of cerebral blood flow, delay time, cerebral blood volume and mean transit time were generated. Delay Time (DT) is a time domain parameter comparable to Tmax, but calculated using a delay and dispersion corrected deconvolution algorithm, as previously described [7, 8]. Artefacts were removed manually where required by a single investigator (JS). The hypoperfused tissue deficit was defined using a threshold of DT >3 seconds, which is comparable to the Tmax+6s threshold utilised in the late window EVT trials [3, 4]. The ischaemic core was defined as the volume of tissue within the hypoperfused region with a relative cerebral blood flow (rCBF) <30% of that in the contralateral hemisphere. Penumbral volume was calculated as the difference between the hypoperfused tissue deficit and the ischaemic core volumes. The criteria for a Target Mismatch profile were based on those used in the DEFUSE-3 study [3]. These included a core volume ≤70ml, penumbra volume ≥15ml and hypoperfusion deficit: core ratio >1.8. The 30% rCBF threshold used in the late window EVT trials was used in the primary analyses [3, 4]. A more conservative estimate of ischaemic core, based on an rCBF threshold of <20% was also assessed, as previously described [9].

Follow-up infarct volumes were measured on follow-up DWI or NCCT in patients without a DWI. Standard planimetric techniques (MRIcron, NITRC, Columbia) with intensity-based region-growing were used to define infarct volumes. Haemorrhagic transformation as well as subarachnoid haemorrhage was assessed on follow-up NCCT. All intracranial haemorrhages

were graded using the Heidelberg Bleeding Classification system [10]. Symptomatic intracranial haemorrhage was defined as any new intracranial haemorrhage, occurring within 36 hours of EVT associated with an increase in NIHSS score of ≥4 points.

### Outcomes

The primary study outcome was good functional status at 90 days, defined as a mRS score of 0–2. Secondary outcomes were follow-up infarct volume, infarct expansion and penumbral salvage volumes as well as haemorrhagic transformation rates.

### Statistical analysis

Statistical analysis was performed using SPSS 22.0 (IBM Corporation, NY). The frequency of clinical and haemorrhagic outcomes in Target Mismatch and Non-Mismatch groups was compared using Pearson Chi Squared tests, or Fisher's Exact tests in instances where >20% of cells had an expected count less than five. Ordinal regression analysis was performed to compare 90-day mRS scores between groups, adjusting for final mTICI score as a co-variate. The effect of penumbral salvage on clinical outcome in Non-Mismatch patients was investigated using a Pearson Chi Squared test. Patients with a premorbid mRS score ≥3 were excluded from functional outcome analysis. Median infarct volumes, infarct expansion and salvage volumes were compared using Mann-Whitney tests.

## Results

### Patients

During the study period, 613 patients were treated with EVT. Forty-one patients with posterior circulation stroke were excluded. Of the patients with anterior circulation stroke, 328 (57%) underwent pre-treatment CTP imaging. In 163 patients, CTP source data were not available to generate objective penumbral maps using MIStar. The remaining 170 patients were included in the final mismatch pattern analysis (Fig 1).

### Perfusion patterns

The majority of patients (151/170, 89%) had objective Target Mismatch on pre-treatment CTP (Fig 1). There were 14 patients with large cores (>70ml) who were labelled as Large Core Non-Mismatch. A further 5 patients did not meet Target Mismatch criteria due to very small perfusion deficits (<15ml)., Application of a more conservative threshold to define ischaemic core (rCBF <20%) resulted in another 7 patients with large cores being re-classified as Target Mismatch (Fig 2).

### Baseline characteristics

Baseline characteristics were similar between patients with and without source CTP images, except for median baseline NIHSS, which was higher in patients included in the final analysis (Table 1). The Large Core Non-Mismatch group were more likely to be male than those in the Target Mismatch group. Other baseline clinical characteristics were similar between groups. Large Core Non-Mismatch patients had significantly larger pre-treatment core volumes and smaller mismatch ratios than the Target Mismatch group, although absolute penumbral volumes were larger in the Large Core Non-Mismatch group (Table 2). All Large Core Non-Mismatch patients had a penumbral volume >15ml and the majority (10/14, 71%) had a mismatch ratio >1.8.

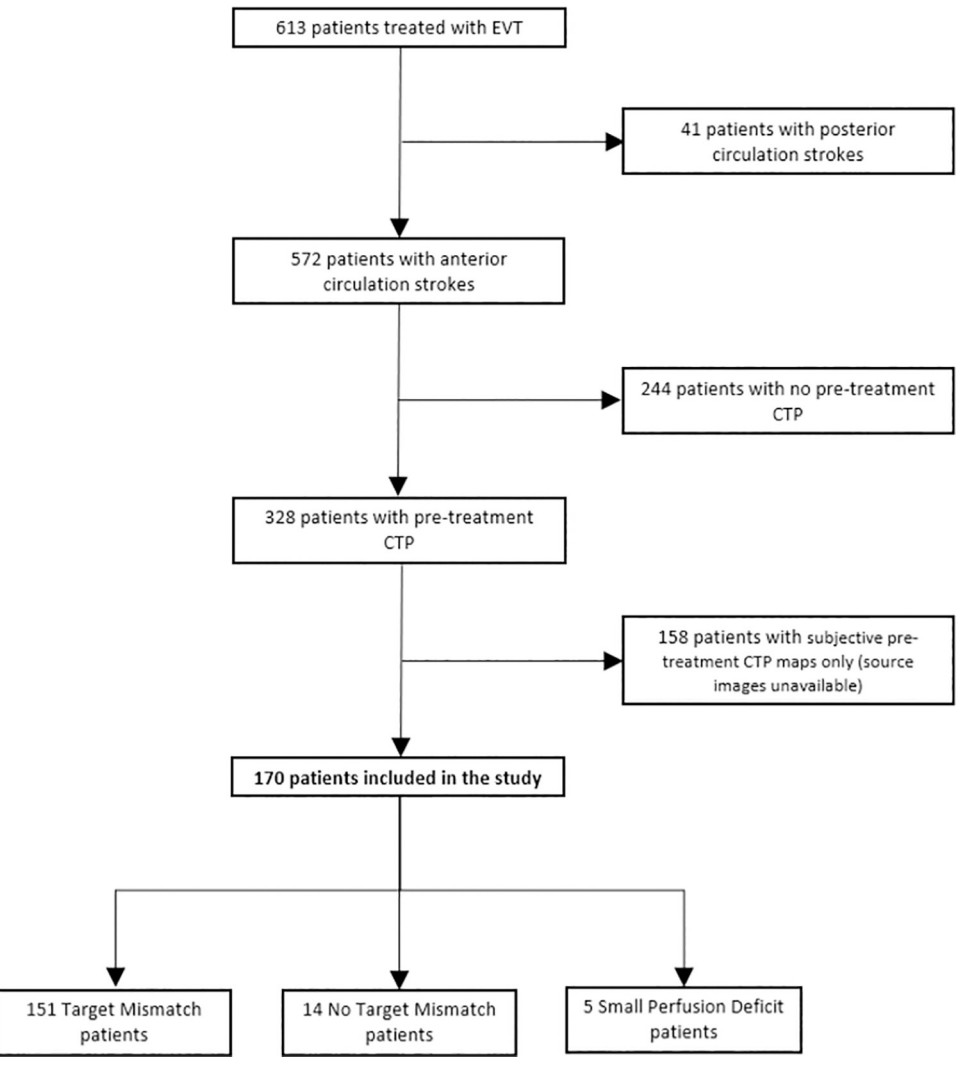

**Fig 1. Consort diagram demonstrating subject inclusion and distribution of patients between groups.** EVT indicates endovascular thrombectomy; CTP, computed tomographic perfusion.

## Clinical outcomes

Day 90 functional outcome data were available for all patients. Eight Target Mismatch patients and one Large Core Non-Mismatch patient with a premorbid mRS score ≥3 were excluded from clinical outcome analysis. Overall, the rate of good (mRS 0–2) and excellent (mRS 0–1) functional outcomes was 53% and 46% respectively. The rate of good functional outcome in the Target Mismatch group (53%) was not different to that in the large core, Large Core Non-Mismatch group (46%, p = 0.629; Table 3). The rate of excellent functional outcomes was higher in the Target Mismatch group (48% compared with 15% in the Large Core Non-Mismatch group, p = 0.023). Target Mismatch was not associated with a shift in 90-day mRS score compared to the Large Core Non-Mismatch group, when adjusted for final mTICI score (Odds Ratio 0.410 [95% CI, 0.149–1.129], p = 0.084; Table 3). Within the Large Core Non-Mismatch group, penumbral salvage was not associated with improved clinical outcomes (Table 4). The rate of good functional outcome in the small perfusion deficit group was 60%.

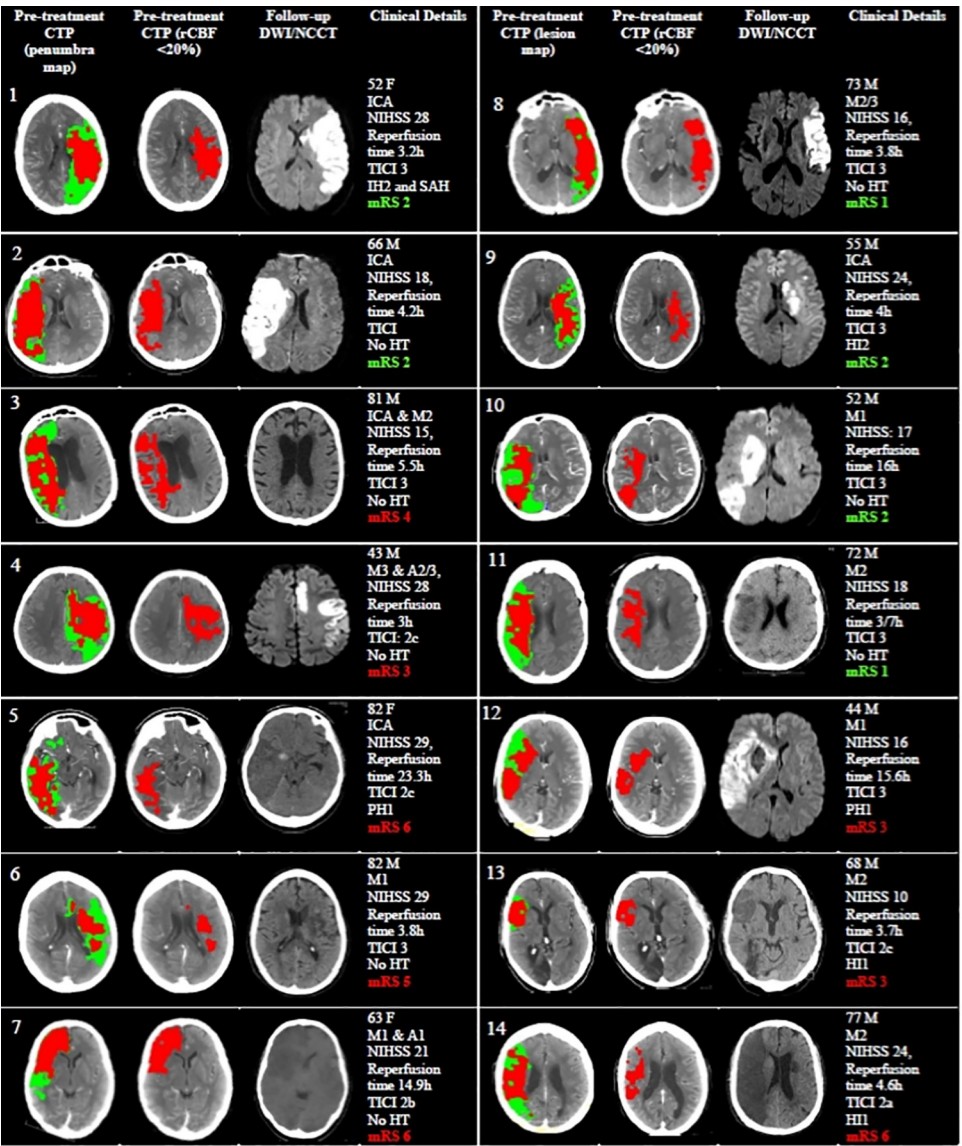

**Fig 2. Imaging results in the 14 Large Core Non-Mismatch patients.** Pre-treatment computed tomographic perfusion (CTP) images were post-processed on MIStar to generate lesion maps (Columns 1, 5). The ischaemic core (red) on the lesion map indicates tissue with relative cerebral blood flow (rCBF) <30%. The surrounding penumbra (green) indicates tissue with delay time (DT) >3 seconds. Sources images were also post-processed to generate ischaemic core maps using rCBF <20% (Columns 2, 6). Patients 1–7 remained as No Target Mismatch using the more conservative rCBF <20% threshold, while patients 8–14 became Target Mismatch. Lesion maps correlated to follow-up infarcts on follow-up diffusion weighted imaging (DWI) and non-contrast computed tomography (NCCT). 6/14 patients achieved good functional outcomes (90-day modified Rankin Scale [mRS] score 0–2), despite having large infarcts. ICA indicates internal carotid artery; NIHSS, National Institute of Health Stroke Scale; mTICI, modified Thrombolysis in Cerebral Infarction; M*n*, middle cerebral artery (*n* segment); HI*n*, haemorrhagic infarction *n*; SAH, subarachnoid haemorrhage; A*n*, anterior cerebral artery (*n* segment); PH1, parenchymal haematoma 1.

## Radiographic outcomes

Follow-up DWI was available in 106 patients (64%) and NCCT was available in a further 58 patients. One patient did not receive follow-up imaging. Median follow-up infarct volume was five-fold larger in the Large Core Non-Mismatch group than the Target Mismatch Group (Table 2). Infarct growth volumes and penumbral salvage ratios were similar between patients

**Table 1. Baseline characteristics in patients with and without objective CTP.**

| | With Objective CTP | Without Objective CTP | P value |
|---|---|---|---|
| **No. of patients (%)** | 170 (30) | 402 (70) | |
| **Clinical** | | | |
| **Age (IQR)** | 73 (63–80) | 77 (66–85) | 0.155 |
| **Female (%)** | 87 (51) | 160 (47) | 0.386 |
| **Baseline NIHSS (IQR)** | 17 (11–22) | 16 (11–20) | 0.038 |
| **Premorbid mRS 0–2 (%)** | 161 (95) | 382 (96) | 0.937 |
| **Pre-EVT Thrombolysis (%)** | 53 (31) | 139 (35) | 0.361 |
| **Baseline Imaging** | | | |
| **ICA occlusion (%)** | 37 (22) | 102 (26) | 0.569 |
| **M1 occlusion (%)** | 96 (56) | 196 (49) | 0.729 |
| **M2 occlusion (%)** | 42 (25) | 98 (25) | 0.958 |
| **Treatment** | | | |
| **Onset to arterial puncture, hrs (IQR)** | 4.4 (3.2–10.6) | 5.2 (3.8–9.8) | 0.056 |
| **Onset to reperfusion, hrs (IQR)** | 5.1 (3.8–10.7) | 5.8 (4.4–11.0) | 0.142 |
| **mTICI flow grade 2c/3 (%)** | 114 (67) | 242 (61) | 0.191 |
| **mTICI flow grade 2b (%)** | 54 (32) | 125 (31) | 0.973 |
| **mTICI flow grade 2a (%)** | 1 (1) | 5 (1) | 0.472 |
| **First pass reperfusion (%)** | 99 (58) | 204 (51) | 0.150 |

All values are median. CTP, computed tomographic perfusion; IQR, interquartile range; NIHSS, National Institutes of Health Stroke Scale; mRS, modified Rankin Scale; EVT, endovascular thrombectomy; ml, millilitres; ICA, internal carotid artery; M1 middle cerebral artery first segment; M2 middle cerebral artery second segment; mTICI, modified Thrombolysis in Cerebral Infarction. Apparent discrepancies in percentages are due to missing data.

with and without Target Mismatch. Median follow-up infarct volume was 6ml (IQR 4ml-63ml) in the 5 small perfusion deficit patients. Infarct growth and penumbral salvage volumes were 6ml (1ml-63ml) and -6ml (-56ml-9ml) respectively in this small group of patients.

Haemorrhagic transformation was common, occurring in 50% of all patients, but the frequency did not differ between Target and Large Core Non-Mismatch patients (Table 2). The single case of symptomatic intracranial haemorrhage occurred in a Large Core Non-Mismatch patient. There were 2 instances of haemorrhagic transformation in the small perfusion deficit patients.

## Treatment window

Most patients included in the study presented in the 'early' (<6 hours from symptom onset) window (111/170, 65%). Within both early and late window groups, there were no significant differences in clinical outcomes between patient with and without Target Mismatch. Within the late window group, the rate of parenchymal haematoma was higher in patients without Target Mismatch. There were no other differences in radiographic outcomes between Target Mismatch and Large Core Non-Mismatch patients in either early or late window groups (Table 5).

## Discussion

This retrospective analysis of perfusion imaging in patients selected as suitable for EVT on clinical and imaging grounds, including non-standardised CTP assessment, demonstrated that the majority of treated patients have Target Mismatch, despite the lack of formal CTP selection at our centres. The rates of good clinical outcome were comparable to those seen in the pivotal

**Table 2. Baseline characteristics.**

| | Target Mismatch | Large Core Non-Mismatch | P value |
|---|---|---|---|
| **No. of patients (%)** | 151 (92) | 14 (8) | |
| **Clinical** | | | |
| **Age (IQR)** | 73 (64–80) | 67 (52–78) | 0.400 |
| **Female (%)** | 82 (54) | 3 (21) | 0.019 |
| **Baseline NIHSS (IQR)** | 17 (10–22) | 20 (16–28) | 0.315 |
| **Premorbid mRS 0–2 (%)** | 143 (95) | 13 (93) | 0.518 |
| **Pre-EVT Thrombolysis (%)** | 46 (31) | 5 (36) | 0.765 |
| **Baseline Imaging** | | | |
| **Time from onset to CTP, hrs (IQR)** | 2.8 (1.5–9.2) | 1.9 (1.3–11.7) | 0.162 |
| **Core volume, ml (IQR)** | 16 (7–32) | 90 (81–125) | <0.001 |
| **Penumbral volume, ml (IQR)** | 82 (59–113) | 102 (88–134) | 0.038 |
| **Mismatch Ratio (IQR)** | 6 (3.7–9.7) | 2 (1.8–2.5) | <0.001 |
| **ICA occlusion (%)** | 30 (20) | 5 (36) | 0.165 |
| **M1 occlusion (%)** | 90 (60) | 5 (36) | 0.084 |
| **M2 occlusion (%)** | 36 (24) | 4 (29) | 0.693 |
| **Left Hemisphere (%)** | 76 (50) | 5 (36) | 0.285 |
| **Right Hemisphere (%)** | 75 (50) | 9 (64) | 0.285 |
| **Treatment** | | | |
| **Onset to arterial puncture, hrs (IQR)** | 4.6 (3.6–10.6) | 3.4 (2.9–14.5) | 0.162 |
| **Onset to reperfusion, hrs (IQR)** | 5.6 (4–11) | 4.1 (3.7–15) | 0.362 |
| **mTICI flow grade 2c/3 (%)** | 98 (65) | 12 (86) | 0.145 |
| **mTICI flow grade 2b (%)** | 51 (34) | 1 (7) | 0.067 |
| **mTICI flow grade 2a (%)** | 2 (1) | 1 (7) | 0.235 |
| **First pass reperfusion (%)** | 91 (62) | 5 (36) | 0.061 |

All values are median. IQR, interquartile range; NIHSS, National Institutes of Health Stroke Scale; mRS, modified Rankin Scale; EVT, endovascular thrombectomy; CTP, computed tomographic perfusion; ml, millilitres; ICA, internal carotid artery; M1 middle cerebral artery first segment; M2 middle cerebral artery second segment; mTICI, modified Thrombolysis in Cerebral Infarction.

trials demonstrating the efficacy of EVT, including those treated more than 6 hours after onset [2–4]. In the small group of patients without Target Mismatch, good outcome rates were similar to those in Target Mismatch patients. In contrast to these good clinical outcomes, radiographic measures of treatment efficacy were markedly different in the Large Core Non-Mismatch patients.

## Selection of EVT candidates

The role of CTP in selection of patients for EVT is controversial. The lack of a uniform approach to CTP in the trials leading to EVT approval have contributed to this. Clinician opinion and comfort with the newest addition to the multimodal CT examination has certainly also contributed to variability in practice patterns as well [11]. Finally, CTP is simply not available at all primary stroke centres and even fewer have the capability to provide post-processed threshold based estimates of core and penumbral volumes using software such as RAPID or MIStar.

At our centres, during the study period, 43% of EVT patients never underwent CTP. Furthermore, in approximately half of the patients who underwent CTP, the imaging available to the clinicians at the time of treatment was limited to qualitative colour maps provided by the hardware vendors. Despite this, the vast majority of the treated patients were in fact excellent

**Table 3. Clinical, radiographic and intracranial haemorrhage outcomes.**

| | DT>3s and rCBF <30% | | | DT>3s and rCBF <20% | | |
|---|---|---|---|---|---|---|
| | Target Mismatch (n = 151) | Large Core Non-Mismatch (n = 14) | P-value | Target Mismatch (n = 158) | Large Core Non-Mismatch (n = 7) | P-value |
| **Clinical Outcomes** | | | | | | |
| **Good Functional Outcome (mRS 0–2) (%)** | 76 (53) | 6 (46) | 0.629 | 80 (54) | 2 (29) | 0.181 |
| Excellent Functional Outcome (mRS 0–1) (%) | 69 (48) | 2 (15) | 0.023 | 71 (48) | 0 (0) | 0.013 |
| **Radiographic Outcomes** | | | | | | |
| **Follow-up Infarct Volume, ml (IQR)** | 16 (8–47) | 104 (25–189) | 0.012 | 17 (8–51) | 155 (16–214) | 0.440 |
| **Infarct Growth, ml (IQR)** | 3 (-5-29) | 9 (-57-77) | 0.402 | 3 (-5-29) | 40 (-65-72) | 1.000 |
| **Penumbral Salvage, ml (IQR)** | 71 (36–100) | 98 (25–131) | 0.576 | 71 (36–101) | 99 (26–205) | 0.699 |
| **Penumbral Salvage Ratio (IQR)** | 0.9 (0.6–1.0) | 0.7 (0.2–1.3) | 0.854 | 0.9 (0.6–1.0) | 0.7 (0.3–1.3) | 0.947 |
| **Haemorrhagic Transformation (%)** | 76 (50) | 7 (50) | 0.981 | 82 (52) | 1 (14) | 0.057 |
| **Parenchymal Haematoma (%)** | 6 (4) | 2 (14) | 0.086 | 7 (4) | 1 (14) | 0.299 |
| **Symptomatic intracranial haemorrhage (%)** | 0 (0) | 1(7) | 0.086 | 0 (0) | 1(14) | 0.043 |

All values are median; DT: delay time; rCBF: relative cerebral blood flow; mRS, modified Rankin Scale; ml, millilitres; IQR, interquartile range.

candidates who met CTP clinical trial criteria applied in both early and late window patients [3, 4, 6]. In addition, the clinical outcomes seen in our patients were generally comparable to those seen in the pivotal clinical trials [2]. While CTP provides valuable prognostic information, its absence does not preclude successful treatment with EVT.

CTP was sometimes obtained in the pivotal EVT early window trials, but the results were not used to select patients for inclusion [12–14]. This is comparable to our own study, where CTP was obtained but not used explicitly to select patients in either the early or late time window. The role of CTP for patient selection in the early window is undefined, but increasingly it is an imaging modality used for diagnosis rather than just prognosis [15].

## EVT in Large Core Non-Mismatch patients

Although, a very limited group in our study, patients with Large Core Non-Mismatch patterns appeared to have good outcome rates that were similar to those in the Target Mismatch group. The rate of good functional outcome in this small group (46%) was higher than that reported in previous retrospective studies of large core patients undergoing EVT in clinical practice (25%—31%) [16, 17]. Our findings similarly contradict previous prospective studies that have demonstrated superior response to EVT in Target Mismatch patients than in patients without Target Mismatch [18–20]. Our results are consistent with a secondary analysis of the MR CLEAN trial, which demonstrated similar treatment effect in patients with and without Target Mismatch [21]. Our findings are ultimately limited by the small sample size of the Large Core

**Table 4. Effect of penumbral salvage on clinical outcomes in Large Core Non-Mismatch patients.**

| | Penumbral Salvage (n = 9) | No Penumbral Salvage (n = 4) | P value |
|---|---|---|---|
| **Good Functional Outcome (mRS 0–2) (%)** | 5 (56) | 1 (25) | 0.343 |
| **Excellent Functional Outcome (mRS 0–1) (%)** | 2 (22) | 0 (0) | 0.462 |

Penumbral salvage was defined as a penumbral salvage ratio >0.5. mRS: modified Rankin Scale.

Table 5. Clinical, radiographic and intracranial haemorrhage outcomes in early and late window.

| | Early Window (<6hr) | | | Late Window (>6hr) | | |
|---|---|---|---|---|---|---|
| | Penumbral (n = 99) | Large Core, Non-Penumbral (n = 10) | P-value | Penumbral (n = 49) | Large Core, Non-Penumbral (n = 4) | P-value |
| **Clinical Outcomes** | | | | | | |
| **Good Functional Outcome (mRS 0–2) (%)** | 55 (59) | 5 (56) | 0.551 | 18 (38) | 1 (25) | 0.521 |
| Excellent Functional Outcome (mRS 0–1) (%) | 51 (55) | 2 (22) | 0.063 | 15 (34) | 0 (0) | 0.236 |
| **Radiographic Outcomes** | | | | | | |
| **Final Infarct Volume, ml (IQR)** | 17 (6–43) | 85 (17–175) | 0.097 | 16 (9–54) | 150 (88–234) | 0.110 |
| **Infarct Growth, ml (IQR)** | 1 (-8-27) | 8 (-64-49) | 0.740 | 5 (-3-34) | 47 (-15-108) | 0.631 |
| **Penumbral Salvage Ratio (IQR)** | 0.9 (0.7–1.0) | 0.7 (0.5–1.4) | 0.740 | 0.9 (0.5–1.0) | 0.6 (0.1–1.2) | 0.631 |
| **Haemorrhagic Transformation (%)** | 52 (53) | 4 (40) | 0.336 | 24 (49) | 3 (75) | 0.320 |
| **Parenchymal Haematoma (%)** | 4 (4) | 0 (0) | 0.677 | 2 (4) | 2 (50) | 0.025 |
| **Symptomatic intracranial haemorrhage (%)** | 0 (0) | 0 (0) | 1.000 | 0 (0) | 1 (25) | 0.077 |

Non-Mismatch group, and the retrospective study design. The effect of EVT in Large Core Non-Mismatch patients remains unknown.

The rate of excellent functional outcome (mRS 0–1) was significantly lower in the Large Core Non-Mismatch group. While we chose the 0–2 mRS threshold as our primary outcome, to be consistent with the landmark EVT trials, the optimal endpoint in stroke trials remains an area of debate [22]. The boundary between an mRS of 2 and 3 can be particularly subjective, which is relevant to unblinded studies such as the present one. The mRS 0–1 outcome is potentially a more rigorous clinical endpoint, as was used in initial trials of thrombolysis in ischaemic stroke [23].

Target Mismatch patients treated with EVT in the present study had small follow-up infarct volumes, and good functional outcomes. Follow-up infarct volumes in the Large Core Non-Mismatch group were substantially larger, however, and in a range generally associated with poor clinical outcomes [24, 25]. The lack of correlation between follow-up infarct volumes and good functional outcome in the Large Core Non-Mismatch group was unexpected, given the extensive evidence demonstrating that infarct volume is highly predictive of functional recovery [24–28]. The reasons for this are unclear, but do emphasise the need for prospective rigorously conducted randomised trials in patients with large cores. It may be that the Large Core Non-Mismatch group represent a highly selected cohort of patients with favourable clinical characteristics. Although not statistically significant, the Large Core Non-Mismatch group tended to be younger and included a high proportion of right hemisphere strokes (9/14; 64%), which may have contributed to better clinical outcomes. The apparent disconnect between radiological and clinical outcomes may also be attributable to the lack of blinding and certification of clinical assessors completing the 90 day mRS score, which is subject to interobserver variability [29]. Certification of examiners, standardised interview structures [30–32] and centralised review of video recorded assessments [33] have all been shown to improve interobserver reliability of the mRS score. These practices were not implemented by the interventional neuroradiologists performing the mRS assessments in our centres, which may have contributed to the apparent discrepancy between mRS scores and follow-up infarct volumes. Another factor that may have contributed to these findings is variability in CTP volumetric estimates between post-processing software. In our study, MIStar was used to provide quantitative estimates of penumbral/core volume. We did not assess other CT Perfusion post-

processing programs, such as RAPID [34]. Variability in CTP volumetric estimates of up to 50% between post -processing software packages have previously been described [35]. It is therefore possible that use of another program may have resulted in either a greater or fewer number of patients meeting the defined standard for Target Mismatch. It may indeed be the case that clinical outcomes were improved by EVT in the Large Core Non-Mismatch patients, through preservation of key eloquent cortical regions critical to functional outcome [36]. For instance, EVT associated salvage of the precentral gyrus from infarction can restore the function of the primary motor cortex, markedly improving functional recovery, even in patients with large infarcts. Nonetheless, measures of follow-up infarct volume and infarct growth are important markers of treatment efficacy. Indeed, these can be viewed as a form of objective performance assessment criteria, which may have been under-utilised in acute ischaemic stroke treatment studies to date. Criteria of this nature are increasingly used in the development of medical devices, including those for other neurovascular conditions [37–39]. The apparent disconnect between clinical and radiographic outcomes will not be resolved without collecting these data.

This study is intrinsically limited by its retrospective design and the absence of a control group not receiving EVT. Conclusions based on our findings are also limited by the very small number of Large Core Non-Mismatch patients.

Careful selection of EVT candidates based on available clinical and imaging data generally results in treatment of patients with Target Mismatch. The effect of treatment in patients without Target Mismatch is unknown but should be assessed in the context of a randomised trial employing objective performance-based evaluation criteria.

## Supporting information

**S1 Fig. Functional outcome (90-day modified Rankin Scale) for patients with and without Target Mismatch.** mRS indicates modified Rankin Scale.
(TIF)

**S1 Data.**
(XLSX)

## Author Contributions

**Conceptualization:** Ken S. Butcher.

**Data curation:** Jayden C. Subramaniam, Andrew Cheung.

**Formal analysis:** Jayden C. Subramaniam, Ken S. Butcher.

**Investigation:** Jayden C. Subramaniam, Andrew Cheung, Nathan Manning, Justin Whitley, Dennis Cordato, Alessandro Zagami, Cecilia Cappelen-Smith, Huiqiao Tian, Chris Levi, Mark Parsons, Ken S. Butcher.

**Methodology:** Jayden C. Subramaniam, Ken S. Butcher.

**Project administration:** Ken S. Butcher.

**Resources:** Ken S. Butcher.

**Supervision:** Ken S. Butcher.

**Writing – original draft:** Jayden C. Subramaniam, Ken S. Butcher.

**Writing – review & editing:** Andrew Cheung, Nathan Manning, Justin Whitley, Dennis Cordato, Alessandro Zagami, Cecilia Cappelen-Smith, Huiqiao Tian, Chris Levi, Mark Parsons, Ken S. Butcher.

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
