## [Decision Letter · Decision Letter 0]

1 Dec 2022

PONE-D-22-28145Most Endovascular Thrombectomy Patients Have Target Mismatch Despite Absence of Formal CT Perfusion Selection Criteria.PLOS ONE

Dear Dr. Butcher,

Thank you for submitting your manuscript to PLOS ONE. After careful consideration, we feel that it has merit but does not fully meet PLOS ONE’s publication criteria as it currently stands. Therefore, we invite you to submit a revised version of the manuscript that addresses the points raised during the review process.

Please tackle the reviewer's comments. 

We look forward to receiving your revised manuscript.

Kind regards,

Nishant Kumar Mishra, MBBS PhD MD

Academic Editor

PLOS ONE

Journal Requirements:

"KB is supported by the NSW Health 18/19 Senior Cardiovascular Scientist award. Grant number: RG192264. Name of funder: NSW Health (https://www.health.nsw.gov.au/)."

Additional Editor Comments:

Please tackle the reviewers' comments.

Reviewers' comments:

Reviewer's Responses to Questions

**Comments to the Author**

1. Is the manuscript technically sound, and do the data support the conclusions?

Reviewer #1: Partly

Reviewer #2: Yes

2. Has the statistical analysis been performed appropriately and rigorously? 

Reviewer #1: Yes

Reviewer #2: I Don't Know

3. Have the authors made all data underlying the findings in their manuscript fully available?

Reviewer #1: Yes

Reviewer #2: Yes

4. Is the manuscript presented in an intelligible fashion and written in standard English?

Reviewer #1: Yes

Reviewer #2: Yes

5. Review Comments to the Author

Reviewer #1: The authors describe an observational case series of EVT patients some of whom underwent CTP prior to treatment and show a high proportion to meet target mismatch criteria.

The Introduction states that late time window EVT is “based on selection using Computed Tomographic Perfusion (CTP) and quantitative post-processing of data using the RAPID software package,” but this is not correct. DAWN used clinical – core mismatch and allowed entry with DWI alone, not requiring CTP (or perfusion imaging of any kind). MRI perfusion was also permitted in DEFUSE-3 (although processed by RAPID).

Could the authors change “final” infarct volume to “follow-up” infarct volume throughout?

The large proportion of patients excluded because CTP source data were lacking is unfortunate as it limits the current series to 170 participants, only 14 of whom lacked a target mismatch profile, and all of these individuals because of core volume >70ml. Since EVT eligibility was determined by clinicians with full access to all imaging, it is unsurprising that the target mismatch population constituted the great majority of cases. The Discussion and Abstract should make it clearer that the population in the study was of patients selected for EVT based on clinical assessment that included all imaging – eg the Discussion opens with a description of the study being of “perfusion imaging patterns in patients undergoing EVT,” whereas this should better read “perfusion imaging in patients selected as suitable for EVT on clinical and imaging grounds.”

Could the failure to meet target mismatch criteria be a consequence of processing by MiStar rather than alternative software (presumably RAPID in all cases)? Discrepancies between CTP volumetric estimates among software are well-recognised and in some cases may be quite large. The median was 90ml so perhaps half the cases likely looked around 70ml if only vendor perfusion maps were available for clinical review. Since the group evidently had a large penumbra volume, therefore good collateral status, this may have been persuasive for clinical selection.

The more surprising finding was that the group without target mismatch had a similar proportion of mRS 0-2 outcomes as the group that had target mismatch. One potential explanation would be a higher proportion of Right hemisphere strokes in this group since clinical outcomes may be better – at least better disguised – in right compared with left hemisphere strokes of similar volume. The text should include a description of hemispheric lateralisation in the groups. It also seems to be an artefact of the mRS 2 threshold since a difference is seen with the mRS 0-1 outcome. The main message perhaps ought to be that mRS 0-1 has significant potential as a more rigorous clinical endpoint (as was seen in the trials of IV thrombolysis, so perhaps not too surprising) as well as the infarct volume that the authors choose to emphasise. The possibility that the lack of mRS training (and perhaps a non-standard approach to obtaining the score) was most relevant to these findings should be given more prominence. Structured interviews significantly lower the observer variability in mRS scoring, and the subjectivity of the mRS 2 versus mRS 3 threshold is greater than for any other boundary in the scale. The mode of scoring is not described, but if a structured interview was not employed then this may also account for the apparent discrepancy.

The penumbral salvage analysis limited to the 14 non-mismatch patients has numbers far too small to allow any meaningful comparison and there should be no conclusion drawn other than this.

Reviewer #2: This is a well written retrospective analysis of EVT outcomes and their relation to baseline perfusion measures. Please find here some specific comments:

Please include information on onset time for your patient cohort into the methods section.

It seems, based on table 2, that the vast majority of the patients in your study were within a 6h time window of known stroke onset. Previous pooled analysis done on MR CLEAN, ESCAPE, REVASCAT, SWIFT-PRIME & EXTEND-IA data have already shown that advanced imaging (CTP) is not necessary for patient inclusion in this situation. It would therefore be a good idea to discuss the added value of your study in relation to the above mentioned papers. In light of this, it would also be interesting to single out those patients who were in the extended time window (beyond 6h) and compare their outcomes (if possible - I expect this will result in an extremely small cohort).

The following two sentences seem to contradict one another, please correct or explain: "A further 5 patients did not meet Target Mismatch criteria due to very small perfusion deficits (<15ml)" and "All No Target Mismatch patients had a penumbral volume >15ml".

Can the authors provide a reference or explanation for the usage of the following criteria to define area of hypoperfusion: "The hypoperfused tissue deficit was defined using a threshold of DT >3 seconds"? It can certainly be argued what the best parameter map or threshold for labelling tissue as hypoperfused might be, however the value which seems to have "established" itself in literature is Tmax > 6s. For readers unfamiliar with MIStar, could you explain the origin of the parameter DT (delay time) - how is it comparable to Tmax?

I recommend including the following reference into your discussion (Borst J, Berkhemer OA, Roos YBWEM, van Bavel E, van Zwam WH, van Oostenbrugge RJ, van Walderveen MAA et al on behalf of the MR CLEAN investigators. Value of Computer Tomographic Perfusion-Based Patient Selection for Intra-Arterial Acute Ischemic Stroke Treatment. Stroke 2015) and discussing your results also in relation to this 2015 paper.

6. PLOS authors have the option to publish the peer review history of their article (what does this mean?). If published, this will include your full peer review and any attached files.

Reviewer #1: No

Reviewer #2: No

---

## [Author Response · Author response to Decision Letter 0]

20 Feb 2023

Reviewer #1: The authors describe an observational case series of EVT patients some of whom underwent CTP prior to treatment and show a high proportion to meet target mismatch criteria.

The Introduction states that late time window EVT is “based on selection using Computed Tomographic Perfusion (CTP) and quantitative post-processing of data using the RAPID software package,” but this is not correct. DAWN used clinical – core mismatch and allowed entry with DWI alone, not requiring CTP (or perfusion imaging of any kind). MRI perfusion was also permitted in DEFUSE-3 (although processed by RAPID).

Response: Yes, we agree that this is true and we had initially over-simplified the evidence in an effort to be succinct. Although our study focuses on the perfusion parameters from DEFUSE-3, there are other proven modalities of identifying patients with salvageable tissue in the late window. We have therefore altered the Introduction lines 56-63 accordingly:

“EVT benefit in the late window (6-24 hours after onset) has been demonstrated in patients with evidence of potentially salvageable tissue, based on advanced Computed Tomographic (CT) or Magnetic Resonance (MR) imaging techniques.3-5 The most common patient selection method utilised is CT Perfusion (CTP) based estimates of acute infarct core and ischaemic penumbra. The latter can be quantified with post-processing of raw image data using the RAPID software package to provide volumes of estimated core and penumbra.3-5”

Could the authors change “final” infarct volume to “follow-up” infarct volume throughout?

Response: We have changed this throughout the manuscript as suggested.

The large proportion of patients excluded because CTP source data were lacking is unfortunate as it limits the current series to 170 participants, only 14 of whom lacked a target mismatch profile, and all of these individuals because of core volume >70ml. Since EVT eligibility was determined by clinicians with full access to all imaging, it is unsurprising that the target mismatch population constituted the great majority of cases. The Discussion and Abstract should make it clearer that the population in the study was of patients selected for EVT based on clinical assessment that included all imaging – eg the Discussion opens with a description of the study being of “perfusion imaging patterns in patients undergoing EVT,” whereas this should better read “perfusion imaging in patients selected as suitable for EVT on clinical and imaging grounds.”

Response: We agree that subjective interpretation of CTP imaging did influence selection of patients for EVT, despite the absence of formal CTP selection criteria. As suggested, we have altered the Discussion lines 281-285, which now read:

“This retrospective analysis of perfusion imaging in patients selected as suitable for EVT on clinical and imaging grounds, including non-standardised CTP assessment, demonstrated that the majority of treated patients have Target Mismatch, despite the lack of formal CTP selection criteria at our centres.”

Could the failure to meet target mismatch criteria be a consequence of processing by MiStar rather than alternative software (presumably RAPID in all cases)? Discrepancies between CTP volumetric estimates among software are well-recognised and in some cases may be quite large. The median was 90ml so perhaps half the cases likely looked around 70ml if only vendor perfusion maps were available for clinical review. Since the group evidently had a large penumbra volume, therefore good collateral status, this may have been persuasive for clinical selection.

Response: We used MIStar in all cases as our standard and yes, it is possible that RAPID (or one of the several other commercially available 3rd party CTP processing packages) would have yielded slightly different quantitative results. Whether this would result in a materially different outcome in our results is something we can only speculate on. Nonetheless, the reviewer’s point is well taken – we chose a standard, but this is not the gold standard (nor is there one in this case). We have added the following to the Discussion lines 357-364 to address this:

“Another factor that may have contributed to these findings is variability in CTP volumetric estimates between post-processing software. In our study, MIStar was used to provide quantitative estimates of penumbral/core volume. We did not assess other CT Perfusion post-processing programs, such as RAPID.30 Variability in CTP volumetric estimates of up to 50% between post-processing software packages have previously been described.31 It is therefore possible that use of another program may have resulted in either a greater or fewer number of patients meeting the defined standard for Target Mismatch.” 

The more surprising finding was that the group without target mismatch had a similar proportion of mRS 0-2 outcomes as the group that had target mismatch. One potential explanation would be a higher proportion of Right hemisphere strokes in this group since clinical outcomes may be better – at least better disguised – in right compared with left hemisphere strokes of similar volume. The text should include a description of hemispheric lateralisation in the groups.

Response: Excellent point – hemispheric lateralisation is indeed a factor that may have contributed to the unexpectedly high rates of good clinical outcome in the Large Core Non- Mismatch Group. We have added a comparison of hemispheric lateralisation between groups to Table 2, which showed no significant difference between Target and Non-Mismatch groups. Nonetheless the Non-Mismatch group is very small and there is a trend to a higher proportion of right hemisphere infarcts in this group, so the point is still one worth considering.

 Target Mismatch Large Core Non-Mismatch P value

Left Hemisphere (%) 76 (50) 5 (36) 0.285

Right Hemisphere (%) 75 (50) 9 (64) 0.285

We have therefore acknowledged the potential impact of hemispheric lateralisation on the high rate of good clinical outcomes in the Large Core Non-Mismatch Group, in the Discussion lines 345-349:

“It may be that the Large Core Non-Mismatch group represent a highly selected cohort of patients with favourable clinical characteristics. Although not statistically significant, the Large Core Non-Mismatch group tended to be younger and included a high proportion of right hemisphere strokes (9/14; 64%), which may have contributed to better clinical outcomes.”

It also seems to be an artefact of the mRS 2 threshold since a difference is seen with the mRS 0-1 outcome. The main message perhaps ought to be that mRS 0-1 has significant potential as a more rigorous clinical endpoint (as was seen in the trials of IV thrombolysis, so perhaps not too surprising) as well as the infarct volume that the authors choose to emphasise. 

Response: We agree that the lack of a consistent finding for mRS 0-1 and 0-2 outcomes is concerning and may be artefactual. We think this partially reflects 1. The very small numbers of patients in the Large Core Non-Mismatch group and 2. The lack of a blinded objective mRS assessment performed by raters prone to bias. The reason we have emphasised the infarct volumes is that they are objective hard endpoints not prone to bias. We have added the reviewer’s points with respect to the mRS 0-1 and 0-2 outcomes to the Discussion lines 330-336:

The rate of excellent functional outcome (mRS 0-1) was significantly lower in the Large Core Non-Mismatch group. While we chose the 0-2 mRS threshold as our primary outcome, to be consistent with the landmark EVT trials, the optimal endpoint in stroke trials remains an area of debate.16 The boundary between an mRS of 2 and 3 can be particularly subjective, which is relevant to unblinded studies such as the present one. The mRS 0-1 outcome is potentially a more rigorous clinical endpoint, as was used in initial trials of thrombolysis in ischaemic stroke.19”

The possibility that the lack of mRS training (and perhaps a non-standard approach to obtaining the score) was most relevant to these findings should be given more prominence. Structured interviews significantly lower the observer variability in mRS scoring, and the subjectivity of the mRS 2 versus mRS 3 threshold is greater than for any other boundary in the scale. The mode of scoring is not described, but if a structured interview was not employed then this may also account for the apparent discrepancy.

Response: We fully agree that the non-standard approach to obtaining mRS scores likely contributed significantly to the apparent disconnect between clinical and radiographic outcomes. This is acknowledged in the Discussion lines 342-346, and we have also added the following sentence to the Discussion lines 350-355, to give more prominence to this possibility.

“These practices were not implemented by the interventional neuroradiologists performing the mRS assessments in our centres, which may have contributed to the apparent discrepancy between mRS scores and follow-up infarct volumes.”

The penumbral salvage analysis limited to the 14 non-mismatch patients has numbers far too small to allow any meaningful comparison and there should be no conclusion drawn other than this.

Response: We agree entirely that the sample size of the Large Core Non-Mismatch group is far too small to draw any meaningful conclusions and do not wish to suggest that our findings support EVT in these patients and have edited the Discussion lines 324-327 accordingly:

“Our findings are ultimately limited by the small sample size of the Large Core Non-Mismatch group, and the retrospective study design. The effect of EVT in Large Core Non-Mismatch patients remains unknown.” 

Reviewer #2: This is a well written retrospective analysis of EVT outcomes and their relation to baseline perfusion measures. Please find here some specific comments:

Please include information on onset time for your patient cohort into the methods section.

Response: We have added the following to Methods lines 121-124:

“The time from symptom onset to CTP was calculated using the contemporaneously maintained clinical database. Patients with a time to CTP greater than 6 hours were classified as late window presenters and were the subject of a secondary analysis.” 

It seems, based on table 2, that the vast majority of the patients in your study were within a 6h time window of known stroke onset. Previous pooled analysis done on MR CLEAN, ESCAPE, REVASCAT, SWIFT-PRIME & EXTEND-IA data have already shown that advanced imaging (CTP) is not necessary for patient inclusion in this situation. It would therefore be a good idea to discuss the added value of your study in relation to the above mentioned papers. In light of this, it would also be interesting to single out those patients who were in the extended time window (beyond 6h) and compare their outcomes (if possible - I expect this will result in an extremely small cohort).

Response: Yes, the majority of patients treated were in the early time window, where the majority of patients were not selected using CT Perfusion. As in our clinical practice, it was sometimes obtained in three of the trials (ESCAPE, SWIFT PRIME and EXTEND IA), but patients were not selected on the basis of these imaging findings. This is a comparable situation to our own study, where CTP was obtained in some cases but not used explicitly to select patients in either the early or late time window. As suggested, we have performed separate analyses for patients presenting in the early and late time window. The results of these analyses can be seen in Table 5 and the Results lines 268-275:

“Treatment Window

Most patients included in the study presented in the ‘early’ (<6 hours from symptom onset) window (111/170, 65%). Within both early and late window groups, there were no significant differences in clinical outcomes between patient with and without Target Mismatch. Within the late window group, the rate of parenchymal haematoma was higher in patients without Target Mismatch. There were no other differences in radiographic outcomes between Target Mismatch and Large Core Non-Mismatch patients in either early or late window groups (Table 5).”

We have also acknowledged the reviewer’s point in the Discussion lines 309-313: 

“CTP was sometimes obtained in the pivotal EVT early window trials, but the results were not used to select patients for inclusion.12-14 This is comparable to our own study, where CTP was obtained but not used explicitly to select patients in either the early or late time window. The role of CTP for patient selection in the early window is undefined, but increasingly it is an imaging modality used for diagnosis rather than prognosis.15” 

The following two sentences seem to contradict one another, please correct or explain: "A further 5 patients did not meet Target Mismatch criteria due to very small perfusion deficits (<15ml)" and "All No Target Mismatch patients had a penumbral volume >15ml".

Response: We can see the source of confusion. The 5 patients with perfusion deficits <15ml represent an entirely separate and unique group (see Figure 1) who technically do meet Target Mismatch due to very small perfusion lesions but carry very different perfusion characteristics and prognosis to the “No Target Mismatch” group, who all had cores >70ml. These two groups of very different patients were therefore separated and labelled “Small Perfusion Deficit” and “No Target Mismatch”. Hence, the two statements above are referring to two different groups. In order to clarify, we have re-labelled the “No Target Mismatch” group as “Large Core Non-Mismatch” throughout the manuscript.

Can the authors provide a reference or explanation for the usage of the following criteria to define area of hypoperfusion: "The hypoperfused tissue deficit was defined using a threshold of DT >3 seconds"? It can certainly be argued what the best parameter map or threshold for labelling tissue as hypoperfused might be, however the value which seems to have "established" itself in literature is Tmax > 6s. For readers unfamiliar with MIStar, could you explain the origin of the parameter DT (delay time) - how is it comparable to Tmax?

Response: Agreed, this is an important distinction. In truth DT3 and Tmax 6 are very similar. We used delay and dispersion corrected Delay Time, which is different than Tmax, calculated with a standard singular value (whether oscillating or block circulant) deconvolution algorithm. This has been described previously and we have now added following statements and the appropriate references in the Methods lines 130-135:

“Delay Time (DT) is a time domain parameter comparable to Tmax, but calculated using a delay and dispersion corrected deconvolution algorithm, as previously described.7, 8 Artefacts were removed manually where required by a single investigator (JS). The hypoperfused tissue deficit was defined using a threshold of DT >3 seconds, which is comparable to the Tmax+6s threshold utilised in the late window EVT trials.3, 4”

I recommend including the following reference into your discussion (Borst J, Berkhemer OA, Roos YBWEM, van Bavel E, van Zwam WH, van Oostenbrugge RJ, van Walderveen MAA et al on behalf of the MR CLEAN investigators. Value of Computer Tomographic Perfusion-Based Patient Selection for Intra-Arterial Acute Ischemic Stroke Treatment. Stroke 2015) and discussing your results also in relation to this 2015 paper.

Response: This is an interesting study showing a potential benefit of EVT in patients without Target Mismatch and further supporting the need for large, randomised trials investigating EVT in these patients. We have added the following sentence acknowledging this study to the Discussion lines 322-324:

“Our results are consistent with a retrospective secondary analysis of the patient cohort from the seminal the MR CLEAN trial, which demonstrated similar treatment effect in patients with and without Target Mismatch.21”

---

## [Decision Letter · Decision Letter 1]

28 Apr 2023

Most Endovascular Thrombectomy Patients Have Target Mismatch Despite Absence of Formal CT Perfusion Selection Criteria.

PONE-D-22-28145R1

Dear Dr. Butcher,

We’re pleased to inform you that your manuscript has been judged scientifically suitable for publication and will be formally accepted for publication once it meets all outstanding technical requirements.

Kind regards,

Eshak I Bahbah

Academic Editor

PLOS ONE

Reviewers' comments:

Reviewer's Responses to Questions

**Comments to the Author**

1. If the authors have adequately addressed your comments raised in a previous round of review and you feel that this manuscript is now acceptable for publication, you may indicate that here to bypass the “Comments to the Author” section, enter your conflict of interest statement in the “Confidential to Editor” section, and submit your "Accept" recommendation.

Reviewer #2: All comments have been addressed

2. Is the manuscript technically sound, and do the data support the conclusions?

Reviewer #2: (No Response)

3. Has the statistical analysis been performed appropriately and rigorously? 

Reviewer #2: (No Response)

4. Have the authors made all data underlying the findings in their manuscript fully available?

Reviewer #2: (No Response)

5. Is the manuscript presented in an intelligible fashion and written in standard English?

Reviewer #2: (No Response)

6. Review Comments to the Author

Reviewer #2: (No Response)

7. PLOS authors have the option to publish the peer review history of their article (what does this mean?). If published, this will include your full peer review and any attached files.

Reviewer #2: **Yes: **Ivana Galinovic MD PhD

---

## [Editor Report · Acceptance letter]

9 May 2023

PONE-D-22-28145R1 

Most Endovascular Thrombectomy Patients Have Target Mismatch Despite Absence of Formal CT Perfusion Selection Criteria. 

Dear Dr. Butcher:

I'm pleased to inform you that your manuscript has been deemed suitable for publication in PLOS ONE. Congratulations! Your manuscript is now with our production department. 

Kind regards, 

on behalf of

Dr. Eshak I Bahbah 

Academic Editor

PLOS ONE